# Network Analysis of Gut Microbial Communities Reveals Key Reason for Quercetin Protects against Colitis

**DOI:** 10.3390/microorganisms12101973

**Published:** 2024-09-29

**Authors:** Yanan Lv, Jing Peng, Xiaoyu Ma, Zeyi Liang, Ghasem Hosseini Salekdeh, Qunhua Ke, Wenxiang Shen, Zuoting Yan, Hongsheng Li, Shengyi Wang, Xuezhi Ding

**Affiliations:** 1Key Laboratory of Veterinary Pharmaceutical Development, Ministry of Agriculture and Rural Affairs, Lanzhou Institute of Husbandry and Pharmaceutical Sciences of Chinese Academy of Agricultural Science, Lanzhou 730050, China; lvyanan01@caas.cn (Y.L.); pj_lucky_7@163.com (J.P.); 18903865563@163.com (X.M.); liangzeyi1@163.com (Z.L.); kqh1298505845@163.com (Q.K.); shane095@foxmail.com (W.S.); yanzuoting@caas.cn (Z.Y.); lihsheng@sina.com (H.L.); 2College of Veterinary Medicine, Inner Mongolia Agricultural University, Hohhot 010010, China; 3Department of Systems Biology, Agricultural Biotechnology Research Institute of Iran, Agricultural Research, Education, and Extension Organization, Karaj 3135933151, Iran; hosseini.salekdeh@mq.edu.au; 4Department of Molecular Sciences, Macquarie University, North Ryde, NSW 2109, Australia; 5College of Veterinary Medicine, Northwest A&F University, Yangling 712100, China

**Keywords:** quercetin, colitis, oxidative stress, inflammation, tight junction, gut microbiota

## Abstract

As one of the most representative natural products among flavonoids, quercetin (QUE) has been reported to exhibit beneficial effects on gut health in recent years. In this study, we utilized a dextran sulfate sodium (DSS)-induced colitis mice model to explore the protective effects and underlying mechanisms of QUE on colitis. Our data demonstrated that QUE oral gavage administration significantly ameliorates the symptoms and histopathological changes associated with colitis. Additionally, the concentration of mucin-2, the number of goblet cells, and the expression of tight junction proteins (such as ZO-1, Occludin, and Claudin-1) were all found to be increased. Furthermore, QUE treatment regulated the levels of inflammatory cytokines and macrophage polarization, as well as the oxidative stress-related pathway (Nrf2/HO-1) and associated enzymes. Additionally, 16S rDNA sequencing revealed that QUE treatment rebalances the alterations in colon microbiota composition (inlcuding *Bacteroidaceae*, *Bacteroides*, and *Odoribacter*) in DSS-induced colitis mice. The analysis of network dynamics reveals a significant correlation between gut microbial communities and microenvironmental factors associated with inflammation and oxidative stress, in conjunction with the previously mentioned findings. Collectively, our results suggest that QUE has the potential to treat colitis by maintaining the mucosal barrier, modulating inflammation, and reducing oxidation stress, which may depend on the reversal of gut microbiota dysbiosis.

## 1. Introduction

Inflammatory bowel diseases (IBD), primarily encompassing Ulcerative Colitis (UC) and Crohn’s disease (CD), represent a chronic and recurrent inflammatory disorder of the gastrointestinal tract. IBD patients present with symptoms such as abdominal pain, diarrhea, and bloody mucopurulent stools [1], although the etiology of the disease remains unclear. A multitude of factors, including intestinal barrier dysfunction, genetic predisposition, immunological abnormalities, inflammation, oxidative stress, and dysbiosis of gut microbiota are considered significant contributors to the pathogenesis of IBD [2,3]. It has been reported that the highest prevalence of IBD occurs in Europe, with UC affecting 505 individuals per 100,000 persons in southeastern Norway and CD affecting 322 individuals per 100,000 persons in Hesse, Germany. In North America, the prevalence of UC is 286.3 per 100,000 persons in Olmsted County, USA, while CD affects 318.5 individuals per 100,000 persons in Nova Scotia, Canada [4]. This disease not only poses a serious threat to the health of humans but also imposes a substantial economic burden on individuals, families, and society at large [5].

The current therapeutic approach for IBD primarily focuses on maintaining remission [6]. Pharmacological agents, such as 5-aminosalicylic acid and corticosteroids, have been shown to be effective in the management of IBD by suppressing inflammation; however, the long-term administration of these medications may result in adverse effects, including the development of conditions like diabetes, hypertension, and osteoporosis, thereby limiting their clinical utility [7]. In recent years, antibody-based therapies, including infliximab, as well as the use of probiotics (such as *Bifidobacterium infantis* and *Saccharomyces boulardii*) [8], have emerged as promising alternatives. Nevertheless, the high cost associated with these treatments remains a significant barrier to their widespread adoption [9]. Consequently, there is an urgent need to develop new, effective, and cost-efficient therapeutic options for the treatment of IBD that exhibit minimal side effects.

For an extended period, plants have been recognized as significant sources of both nutrient and non-nutrient components beneficial to human health. Consequently, increasing the proportion of plant-based foods in the diet has been regarded as a longstanding nutritional priority in the fight against chronic disease. Flavonoids, a class of plant secondary metabolites, have been shown to enhance gut health by modulating gut microbiota, immune cells, intestinal barrier functions, signaling pathways, inflammation, and oxidative stress [10]. Several studies have investigated the protective and therapeutic effects of dietary flavonoids on colitis induced by dextran sulfate sodium (DSS), including the compounds apigenin [11], genistein [12], fisetin [13], naringin [14], and so on. Furthermore, an increasing number of human intervention studies support the plausibility of flavonoids as a treatment or adjuvant therapy for IBD [10]. Thus, studying the effects of flavonoids holds profound significance for the prevention and treatment of IBD. 

Quercetin (QUE), the most prevalent naturally occurring flavonoid found in various commonly consumed food items, is also utilized as a commercial dietary supplement and incorporated into functional foods [15]. Statistically, the daily human intake of QUE ranges from approximately 20 to 100 mg across different populations [16]. Numerous scientific studies have documented a wide array of biological effects attributed to QUE, including antioxidant, anti-inflammatory, immunoprotective, and potential anti-carcinogenic properties [17]. Specifically, QUE has been shown to inhibit oxidative stress induced by negative stimuli by the overproduction of reactive oxygen species (ROS) and the level of the lipid peroxidation end-product malondialdehyde (MDA) [18]. Furthermore, QUE can cause an inflammatory response by suppressing leucocyte recruitment and decreasing the pro-inflammatory cytokines and chemokine production to ameliorate tissue injury [15,18]. Additionally, QUE modulates several signaling pathways, including Wnt/β-catenin and MAPK/ERK1/2 MAPK/ERK1/2, contributing to anti-cancer effects on cancer cells and tumors [19]. Recent research has demonstrated that QUE plays a crucial role in protecting gut health and maintaining the balance of the gut microbiota. Specifically, QUE has been associated with a reduction in atherosclerotic lesions, which is characterized by a decrease in the abundance of *Verrucomicrobia*, alongside an increase in the abundance of *Actinobacteria*, *Cyanobacteria*, and *Firmicutes* in mice models [20]. Porras et al. reported that QUE administration could significantly attenuate lipoperoxidation-dependent TLR-4 pathway activation by improving gut dysbiosis in mice subjected to a high-fat diet [21]. Feng et al. further illustrated that dietary QUE supplementation ameliorates intestinal inflammation and improves intestinal functions via modulating gut microbiota composition, particularly the increased population of SCFA-producing bacteria [22]. Li et al. demonstrated that the administration of QUE can increase sphingosine levels in the prefrontal cortex (PFC). This process further regulates neuroceramide and 1-phosphosphingosine levels through the action of the gut microbiota, ultimately enhancing sphingolipid metabolism and alleviating behaviors associated with depression [23]. Several studies have indicated that QUE exhibits effects by modulating the cGAS-STING pathway to regulate macrophage polarization, and this modulation subsequently improves the structure and function of the intestinal barrier and alleviates colitis induced by DSS [24]. Wang et al. found that QUE also enhances the integrity damage of the barrier compromised by DSS through the activation of the hydrocarbon receptor [25]. Recent studies have indicated that dietary QUE enhances colonic microbial diversity and mitigates the severity of colitis in mice infected with Citrobacter rodentium [26]. Despite the extensive evidence suggesting that QUE may serve as a beneficial therapeutic agent for the treatment of colitis, the underlying method by which QUE ameliorates colitis remains poorly understood.

Although it is well-established that the gut microbiota plays a significant role in the pharmacological effects of QUE, current available results are limited concerning the action of QUE on gut dysbiosis. Against this background, the purpose of this study is to explore the key flora that affect QUE’s pharmacological action and the interaction between the key flora and microenvironmental factors, so as to provide a theoretical basis for further elucidation of the mechanism of QUE’s action and possible regulatory mechanisms involving the gut microbiota in the treatment of colitis. The results from this study will elucidate the effect of QUE on colitis and provide more mechanisms by which QUE relieves colitis.

## 2. Materials and Methods

### 2.1. Animals and Experimental Design

Forty male C57BL6/J mice (aged 6–8 weeks and weighing 20 ± 2 g) were obtained from the Lanzhou Veterinary Research Institute of the Chinese Academy of Agricultural Sciences (Lanzhou, China) and raised at a constant temperature (25 ± 2 °C) and photoperiod (12/12 h light/dark period) in this study. The mice were randomly separated into four groups: control, QUE, DSS, and DSS-QUE (n = 8). The experiments were performed continually for 2 weeks. The mice from the QUE and DSS-QUE groups received QUE (Sigma, Erlengrund, Germany) treatment by intragastric gavage from day 1 to day 14 (50 mg/kg BW) daily. The mice from the control and DSS groups were administered the same amount of normal saline by gavage every day. On day 8, the mice in the DSS and DSS-QUE groups were given 3% DSS (*w*/*v*) (Yeasen, Shanghai, China) distilled water for an additional 7 days. Following the experiment, blood was collected via retroorbital sinus puncture, maintained at room temperature for one hour, and subsequently centrifuged at 3500 rpm for 15 min to isolate the serum, which was then stored at −20 °C. The mice were euthanized using chloral hydrate, and both fecal matter and colon tissue were collected and promptly divided into several sections. One section of the colon tissue was designated for histological analysis, while the remaining colon tissue and fecal matter were transferred to liquid nitrogen for preservation. Animal welfare and experimental procedures for this study were followed in accordance with the ethical regulations of the Lanzhou Institute of Husbandry and Pharmaceutical Sciences of the Chinese Academy of Agricultural Science (SCXR (gan) 2020-000).

### 2.2. Assessment of Colitis

Mice were weighed daily until sacrifice. The disease activity index (DAI) of mice was recorded by evaluating the clinical symptoms (such as body weight loss, stool consistency, and gross blood) of the mice every day. When the treatment was finished, mice were all sacrificed by cervical dislocation, and the colon tissues were excised and measured. For histological analysis, the formalin-fixed colon tissues were embedded, cut into 5 μm sections, and hematoxylin and eosin (H&E) stained. Sections were visualized under a digital slice scanner (3DHISTECH, Budapest, Hungary). After that, the valuation system of pathological scores in the colon tissue was undertaken according to the previously reported method [27].

### 2.3. Levels of DAO and D-LA in Serum

The levels of DAO and D-LA in the serum were quantified by commercially available ELISA kits (MEIMIAN, Yancheng, China). The detection values of the DAO and D-LA in serum were determined using the Multiskan Spectrum Microplate Spectrophotometer (Biotek Epoch, Winooski, VT, USA).

### 2.4. Level of Cytokines in Colon Tissues

The freshly excised colon was rinsed, homogenized in tissue lysis buffer, and then centrifuged at 10,000× *g* at 4 °C for 15 min, and the supernatant was collected. The levels of IL-1β, TNF-α, IL-6, and IL-10 in the supernatant were quantified by commercially available ELISA kits (MEIMIAN, Yancheng, China). The protein concentration of each sample was measured by the bicinchoninic acid (BCA) protein assay kit (Jiancheng, Nanjing, China). Finally, the detection values of the cytokines and BCA were determined using the Multiskan Spectrum Microplate Spectrophotometer (Biotek Epoch, Winooski, VT, USA).

### 2.5. Level of MUC2 in Colon Tissues

The level of MUC2 in colon tissues was detected by an ELISA kit (MEIMIAN, Yancheng, China) in the supernatants of freshly isolated pieces of colon tissues homogenized with potassium phosphate buffer and centrifuged at 10,000× *g* at 4 °C for 15 min. Finally, the detection value of the MUC2 was determined using the Multiskan Spectrum Microplate Spectrophotometer (Biotek Epoch, Winooski, VT, USA).

### 2.6. Antioxidant Enzyme Activity Determination

This supernatant for antioxidant enzyme activity determination is the same as the one used in the above experiment. The changes in colonic MDA, CAT activity, GSH, and SOD activity were measured by the corresponding kits (Jiancheng, Nanjing, China). Finally, the Multiskan Spectrum Microplate Spectrophotometer (Biotek Epoch, Winooski, VT, USA) was used to determine the detection values of MDA, CAT, and SOD.

### 2.7. Alcian Blue and PAS Staining

Alcian blue staining was used to investigate the distribution of mucin in the colon as per the method of Steedman, and PAS staining was used to detect the number of goblet cells as previously described [7]. Sections were visualized under a biological microscope (Motic BA210, Xiamen, China).

### 2.8. RNA Extraction and qRT-PCR

Total RNA was extracted using TRIzol reagent (Vazyme, Nanjing, China) and quantified using a BioSpectrometer (Eppendorf, Hamburg, Germany). A RevertAid First Strand cDNA Synthesis Kit (Thermo Fisher Scientific Inc., Waltham, MA, USA) was used to reverse-transcribe mRNA to cDNA. qRT-PCR was performed with ChamQ Universal SYBR qPCR Master Mix (Vazyme, Nanjing, China) and detected with AppliedBiosystems QuantStudio5 (Thermo Fisher Scientific, Inc., Waltham, MA, USA). Primer sequences used for qRT-PCR (Sangon Biotech, Shangahi, China) were as follows: mouse GAPDH, GGTTGTCTCCTGCGACTTCA (forward), and TGGTCCAGGGTTTCTTACTCC (reverse); mouse ZO-1, GACCAGTACCCGCGAAG (forward), and CAGAGGAGGGACAACTGC (reverse); mouse Claudin-1, TATCGGAACTGTGGTAGA (forward), and CAGGGAAGATGGTAAGG (reverse); mouse Occludin, TGAAAGTCCACCTCCTTACAGA (forward), and CCGGATAAAAAGAGTACGCTGG (reverse). The different gene expression levels of each group were calculated using the comparative Ct (2^−ΔΔCT^) method. 

### 2.9. Immunohistochemistry

Immunohistochemistry was performed for the expression and distribution of TJ proteins (such as ZO-1, Claudin-1, and Occludin) in the colon. For details, colon tissue sections were dewaxed, dehydrated, and rehydrated, and antigen retrieval was undertaken in a graded ethanol series and distilled water, and then treated with 3.0% hydrogen peroxide for 10 min to block the activity of endogenous peroxidase. After blocking, the primary antibodies against ZO-1 (Servicebio, Wuhan, China), Claudin-1 (ZENBIO, Chengdu, China), or Occludin (Abcam, Cambridge, UK) were incubated separately at 4 °C overnight, and then biotinylated secondary antibodies (ZSGB-BIO, Beijing, China) were used for detection. The staining of the sections was performed using streptavidin-HRP conjugates (ZSGB-BIO, Beijing, China) for ZO-1, Claudin-1, or Occludin. Cleaning for each step was undertaken with PBS after completion. Peroxidase oxidation of diaminobenzidine substrate (DAB; Servicebio, Wuhan, China) was used to visualize the immunospecific reactivity. Subsequently, using hematoxylin counterstain, alcohol, and xylene dehydration, glass slides were mounted and covered with a coverslip. Finally, sections were examined and photographed utilizing a digital triocular microcamera system (Motic BA400Digita, Xiamen, China). 

### 2.10. Immunofluorescence

For immunofluorescence staining, the paraffin-embedded colonic sections were deparaffinized, rehydrated, and washed in 1% PBS-Tween 20, treated with 3% hydrogen peroxide, blocked in 1% BSA for 60 min, incubated with primary antibodies Fluor^®^ 488 rat monoclonal to F4/80 (Abcam, Cambridge, UK), CD86 rabbit polyclonal antibody (Proteintech, Rosemont, IL, USA), CoraLite^®^594-Conjugated CD206 mouse monoclonal antibody (Proteintech, Rosemont, IL, USA), and then Nrf2 mouse monoclonal antibody (Proteintech, Rosemont, IL, USA) and HO-1 rabbit polyclonal antibody (Proteintech, Rosemont, IL, USA) overnight at 4 °C, and the primary antibody was detected with the secondary antibody Alexa Fluor^®^ 488 goat anti-mouse antibody (Abcam, Cambridge, UK), Alexa Fluor^®^ 488 goat anti-rabbit antibody (Abcam, Cambridge, UK), and Alexa Fluor^®^ 647 donkey anti-rabbit antibody (Abcam, Cambridge, UK) for 1 h at 37 °C in the dark. Finally, DAPI (Abcam, Cambridge, UK) was used to stain the nucleus. Cleaning was undertaken for each step with 1% PBS-Tween 20 after completion [28]. Images were visualized by a laser scanning confocal microscope (ZEISS Axio Observer A1, Oberkochen, Germany).

### 2.11. Gut Microbiota Analysis

The cetyltrimethylammonium bromide (CTAB) method was used to extract the DNA from mice fecal samples. A NanoDrop 2000 spectrophotometer (ThermoFisher Scientific, Wilmington, DE, USA) was used to determine the concentration of DNA, and 1% agarose gel electrophoresis was applied to detect the relative purity. After that, the primers 338F (ACTCCTACGGGAGGCAGCA) and 806R (GGACTACHVGGGTWTCTAAT) were used to amplify the V3–V4 region of 16S rDNA for 30 cycles by an Illumina Novaseq 6000 (Illumina, San Diego, CA, USA). Then, the sequencing of products was purified and mixed. The sequencing data were filtered, trimmed, and then used for ASVs and taxonomic analysis.

### 2.12. Statistical Analyses

The data are presented as the mean ± SEM. GraphPad Prism 6 was used to analyze the data. The statistical significance of the differences between each group was determined using a one-way analysis of variance (ANOVA) followed by a post hoc test (LSD) was used. A *p* value of less than 0.05 was considered significant and less than 0.01 was considered highly significant.

## 3. Results

### 3.1. QUE Treatment Improved the Symptoms of DSS-Induced Colitis

The chemical structure of QUE is illustrated in Figure 1A. According to a previous study, we established a colitis model in mice that closely resembles human IBD through the oral administration of DSS [29]. Considering that colitis is a chronic disease, we selected the dosage of QUE based on previous research [30,31]. The experiment was conducted according to the time axis shown in Figure 1B. The changes in the body weight and DAI score were monitored daily throughout the duration of the experiment. As can be seen from Figure 1C, after drinking 3% DSS for 7 days, the body weight of mice remarkably dropped compared to the control group (*p* < 0.01), while QUE administration mitigated this decline. Furthermore, QUE treatment effectively alleviated the symptoms of diarrhea and hematochezia, as evidenced by the DAI scores of the mice (*p* < 0.01) (Figure 1D). The average length of colon in the control group was measured at 8.09 ± 0.28 cm. In addition, the colon in the control group exhibited a normal reddish coloration, and the stool was granular in texture. In contrast, the colon length of the DSS-treated mice was recorded at 5.70 ± 0.63 cm, characterized by a dark red coloration, swelling, and bleeding. A reduction of 29.54% in colon length was observed in the DSS-treated group compared to the control group; however, QUE administration effectively mitigated the colon shortening process (Figure 1E,F). Diamine oxidase (DAO) and D-lactic acid (D-LA) are important indicators of intestinal functional status [32,33]. As illustrated in Figure 1G,H, the levels of DAO and D-LA were significantly elevated in the DSS-induced mice compared to those in the control group. Notably, QUE administration can significantly reverse the upregulation of both DAO and D-LA. 

### 3.2. QUE Alleviated Colonic Tissue Damage Caused by DSS and Protected the Intestinal Barrier

In this study, histopathological evaluation was used to reveal the epithelial layer disruptions, goblet cell loss, and inflammatory cell infiltration in the submucosa of the DSS group (Figure 2A). In contrast, QUE administration resulted in relatively intact crypt structures, reduced distortion of the epithelium, and diminished infiltration of inflammatory cells, resulting in a significant decrease in histopathological scores (Figure 2B). In order to further verify the effect of QUE on intestinal epithelial barrier function, we examined TJ proteins in colonic tissue using quantitative reverse transcription polymerase chain reaction (qRT-PCR) and immunohistochemistry. Compared to the control group, the expression levels of ZO-1, Claudin-1, and Occludin were markedly decreased in the colon tissues of DSS-induced colitis mice, but QUE treatment significantly enhanced the expression of these TJ proteins (Figure 2C,D). To gain a deeper understanding of the effects of QUE on mucosal barrier functions, we subsequently examined the colonic mucus layer and goblet cells. As shown in Figure 3A–D, Alcian blue and periodic acid–Schiff (PAS) staining revealed that the number of mucus-producing goblet cells was severely diminished, and substantial amounts of Mucin2 (MUC2) were lost in the colon of DSS-induced colitis mice compared to those in the control group. Nevertheless, QUE administration significantly mitigated the reduction of goblet cells and the destruction of the mucosal layer induced by DSS.

### 3.3. QUE Could Reduce DSS-Induced Inflammatory Cytokine Secretion and Inhibit Macrophage Polarization

Considerable research has demonstrated that the secretion of pro-inflammatory cytokines and the infiltration of macrophages play a crucial role in the development and maintenance of IBD [34]. In this study, we further examined the levels of inflammatory cytokines and macrophage polarization in the colon to assess the protective effects of QUE. As illustrated in Figure 4A–C, QUE treatment significantly reduced the levels of inflammatory cytokines (such as interleukin-1 beta (IL-1β), tumor necrosis factor alpha (TNF-α), interleukin-6 (IL-6)) in the colon of mice. Conversely, the level of the anti-inflammatory cytokine IL-10 was decreased in the DSS-induced colitis mice colon compared with the control group, but it increased after QUE treatment (Figure 4D). Given that the production of pro-inflammatory chemokines can recruit macrophages into the mucosa of the colon, we subsequently examined the effects of QUE administration on macrophages of the colon. Immunofluorescence staining results revealed a significant infiltration of M1 macrophages (CD86) at the lesion sites in the colon of DSS-treated groups, whereas QUE treatment resulted in a reduction in the number of macrophages present in the colon. Meanwhile, we also observed the quantity of M2 macrophages (CD206) and found that QUE treatment could recover the M2 macrophage numbers that had been diminished by DSS treatment (Figure 4E). Collectively, our findings demonstrate that QUE can mitigate the DSS-induced inflammation response in the colon.

### 3.4. QUE Ameliorates DSS-Induced Oxidative Stress in the Colon of Mice

To evaluate the influence of QUE on DSS-induced oxidative stress in mice, we examined several key regulators involved in oxidative stress in the colon, such as MDA levels, CAT activity, glutathione (GSH) levels, and superoxide dismutase (SOD) activity. Our results showed that the levels of MDA were significantly increased in mice given water with 3% DSS compared to mice in the control group, which could be improved by QUE treatment (Figure 5A). In addition, DSS administration resulted in a decrease in both catalase (CAT) activity and GSH levels compared with the control and QUE groups. Comparing the DSS-QUE group with the DSS group, the former resulted in significantly higher CAT activity and GSH levels (*p* < 0.05) (Figure 5B,C), but showed no significant changes in SOD activity (Figure 5D). Nrf2 is a major detoxification regulator that has been shown to activate the transcription of genes that code for antioxidant enzymes [35]. Therefore, we investigated the role of Nrf2 in mediating the protective effects of QUE against DSS-induced colitis. As shown in Figure 5E,F, the expression levels of Nrf2 and HO-1 were significantly reduced in mice exposed to DSS. Consistently, QUE treatment not only restored but also further enhanced the expression levels of Nrf2 and HO-1. 

### 3.5. Regulation of Gut Microbiota Composition by QUE

Gut microbiota dysbiosis has been widely reported in patients and animal models with IBD [36]. Therefore, the composition of the gut microbiota in mice was investigated using 16S rRNA amplicon sequencing. As shown in Figure 6A, the Simpson and Shannon indexes were used to evaluate α-diversity. After QUE treatment, both the Simpson and Shannon indexes decreased compared to the microbiota from the DSS group. β-Diversity was assessed through the weighted UniFrac distance principal coordinates analysis (PCoA). The results showed that DSS treatment significantly changed the gut microbiota composition, while QUE administration could alleviate and shift the gut microbiota composition induced by DSS treatment (Figure 6B). Upon analyzing the composition of the gut microbiota, we identified 577 universal amplicon sequence variants (ASVs) from a total of 35,192 ASVs across all samples. There were 8501, 7849, 7238, and 5721 unique ASVs in the control, QUE, DSS, and DSS-QUE groups, respectively (Figure 6C). The diversity of gut microbiota among different groups was further analyzed by linear discriminant analysis effect size (LEfSe). Following the DSS challenge, there was a significant increase in *Bacteroidiaceae* at the family level and *Bacteroides* at the genus level, with these shifts being reversed by the QUE supplementation (Figure 6D,E). Histograms of linear discriminant analysis (LDA) scores were generated to identify statistically significant biomarkers and show the dominant microorganisms within each group. A score histogram of LDA was used to identify statistically significant biomarkers and show the dominant microorganisms in each group (Figure 6F,G). Notably, *Verrucomicrobiaceae*, *Verrucomicrobia*, *Akkermansia*, *Verrucomicrobiae*, and *Verrucomicrobiales* were identified as the dominant bacteria in the QUE group; *Bacteroidaceae* and *Bacteroides* were dominant in the DSS group; *Clostridiales* and *Clostridia* were the dominant microbes in the DSS-QUE group (Figure 6G) (LSD ≥ 3). In addition, KEGG analysis and MetaCyc analysis are used to better understand the potential function of colonic microbiota. As shown in Figure 7A,B, our analysis revealed that amino acid metabolism and biosynthesis, carbohydrate metabolism and biosynthesis, cofactors and vitamins metabolism and biosynthesis, and nucleotide metabolism and biosynthesis were the main pathways identified in the metabolic analysis.

### 3.6. The Correlations between Gut Microbes and Micro-Environmental Factors

As shown in Figure 8, the relationship between the 23 most abundant genera and various parameters, including the DAI, pathological score, colon length, levels of inflammatory cytokines, and several key regulators involved in oxidative stress, were analyzed. *Bacteroides* and *Bacteroidaceae* abundance exhibited a significant positive correlation with TNF-α, IL-6, IL-1β, MDA, DAO, DAI score, and pathological score. Conversely, these genera exhibited a significant negative correlation with body weight, colon length, goblet cell numbers per crypt, and the levels of SOD, CAT, GSH, and IL-10 in the colon. In addition, *Lactobacillaceae*, *Lactobacillus*, and *Lactobacillales* abundance, which decreased in the DSS and DSS-QUE group, were significantly negatively correlated with IL-6, MDA, DAI score, and pathological score, while showing a significant positive correlation with body weight and colon length. Interestingly, the abundance of *Akkermansia* exhibited a significant positive correlation with the MUC2 levels in the colon. Collectively, these findings indicate that QUE treatment alleviates intestinal inflammation and oxidative stress, potentially through the increase in the abundance of *Lactobacillaceae*, *Lactobacillus*, and *Lactobacillales*, alongside a decrease in the abundance of *Bacteroides* and *Bacteroidaceae*.

## 4. Discussion

IBD is a chronic and recurrent inflammatory condition of the gastrointestinal tract that significantly impacts the quality of life of approximately 4 million individuals [37]. Currently, traditional treatment options for patients with IBD include salicylate, steroids, immunosuppressants, and anti-TNF-α agents. However, the use of these medicines not only increases susceptibility to infection, but also causes a variety of side effects during the course of IBD treatment [38,39,40,41]. Consequently, exploring a novel therapeutic and preventive approach for IBD treatment is urgent and necessary.

QUE, a kind of flavonoid, has been observed to confer significant therapeutic potential against intestinal disease [17,21,22]. However, the therapeutic effect and mechanism of QUE in IBD still remain incompletely clarified. In the current study, we found that some clinical symptoms induced by DSS were significantly alleviated by QUE treatment. Notably, QUE treatment resulted in the recovery of weight loss, reduction of diarrhea, and alleviation of hematochezia, which collectively led to a marked decrease in the DAI score. In addition, QUE treatment significantly reduced the pathological score and increased colon length, providing further evidence that QUE can substantially ameliorate histopathological damage to the colon and alleviate the symptoms of colitis induced by DSS.

Composed of a mucous layer and an epithelial cell layer, the mucosal barrier is regarded as the first line of defense against a hostile environment. It serves to prevent the invasion of intestinal bacterial toxins and other exogenous substances into the intestinal tissues while facilitating nutrient absorption. Previous studies showed that MUC2 knockout mice could spontaneously develop colitis and have less resistance to the invasion of enteral pathogens [42,43]. In our study, we observed that QUE effectively maintained the integrity of the colonic mucous layer by increasing the number of mucus-producing goblet cells and the concentration of MUC2. It has been reported that the increased permeability of the TJ can accelerate the development of inflammatory diseases, in addition to causing damage to the mucous layer [44]. As a key component of the intestinal mucosal barrier to prevent the spread of pathogens and harmful antigens across the epithelium, TJ proteins seal the space between adjacent intestinal epithelial cells and play an important role in maintaining intestinal permeability in IBD [45]. As reported, TJ proteins such as ZO-1 have been found to be significantly decreased in patients with UC [46], and alterations in TJ integrity have been shown to precede the exacerbation of inflammation in various mouse models [47,48]. In this study, we found that the mice induced by DSS consistently exhibited compromised intestinal epithelial barriers and inflammatory responses. However, QUE treatment restored the protein expression of TJ proteins, including ZO-1, Occludin, and Claudin-1, thereby demonstrating the protective effect of QUE on epithelial barrier integrity, which aligns with findings from previous studies [7]. 

Despite the mucosa serving as a protective barrier in the intestine, the ingestion of materials and microbial pathogens can induce oxidative stress injury and inflammatory responses by stimulating the epithelium and immune/inflammatory cells [49]. As reported, an inflammatory microenvironment could affect epithelial barrier properties and mucosal homeostasis through both direct and indirect mechanisms that alter the structure and function of epithelial intercellular junctions [50]. Mononuclear phagocytes and epithelial cells in the intestine can drive inflammatory cytokines, including IL-1β, TNF-a, and IL-6, which are involved in the occurrence and progression of colitis [51]. Consistent with a previous study [29], we found that IL-1β, TNF-α, and IL-6 were significantly increased in the DSS group compared with the control group. However, administration of QUE resulted in a reduction of these pro-inflammatory cytokines. Additionally, we observed a significant decrease in the levels of IL-10 in the colon of the DSS group, while QUE treatment led to an increase in IL-10 levels. IL-10 is an anti-inflammatory cytokine produced by various cell types, including B and T lymphocytes, macrophages, and dendritic cells. It has been demonstrated to play a protective role in maintaining gut homeostasis and preventing colitis [52]. Talero et al. demonstrated that mice with a knockout IL-10 gene can spontaneously develop colitis at 6 weeks. IL-10 gene knockout is one of the earliest ways employed to establish colitis models for animals [53], indicating that IL-10 serves as a critical inflammatory marker in the context of colitis. As a primary contributor to potentially pathological inflammatory processes, macrophages can rapidly produce large amounts of inflammatory cytokines in response to danger signals [54]. It is reported that a number of macrophages contribute to the development and perpetuation of colonic inflammation [35]. In this study, F4/80-positive staining was used to identify macrophages. Immunofluorescence results indicated that QUE treatment could reduce the percentage of M1 macrophages while increasing the percentage of M2 macrophages after DSS induction. M1 macrophages produce pro-inflammatory cytokines and nitric oxide, causing mucosal damage, whereas M2 macrophages secrete anti-inflammatory factors that promote mucosal repair [55]. Zhu et al. reported that the percentage of M1 macrophages was increased, while the percentage of M2 macrophages was decreased in colitis [56]. Given that macrophage polarization plays a crucial role in the process of colitis, we further studied the effects of QUE on the phenotype and function of macrophages. Our findings suggest that QUE administration could inhibit M1 macrophages while promoting M2 macrophages, indicating that maintaining a balance between M1 and M2 macrophages may be one of the possible mechanisms by which QUE decreases the severity of acute DSS-induced colitis. Furthermore, our results demonstrated that QUE not only corrected the expression of these inflammatory cytokines but also suppressed macrophage infiltration and polarization during colitis. This supports previous studies indicating that cytokines and cytokines networks play an important role in maintaining gut health by modulating the function of the epithelial barrier, the immune response, and tissue integrity [57].

The pathogenesis of various gastrointestinal diseases, such as peptic ulcers, gastrointestinal cancers, and IBD, is partially linked to oxidative stress, and deficiencies in mucosal antioxidant defenses, which are significant contributors to IBD. In mice models of UC, inflammatory responses caused by DSS primarily occur through the activation of the IκBα and NF-κB pathways, which are stimulated by ROS. It has been reported that DSS treatment can increase the sulfate load of cells and further induce ROS production, leading to the activation of inflammatory response. Similarly, a sulfur-rich diet in human UC can induce inflammation mediated by ROS [49]. Furthermore, ROS enhances the immune response in IBD through inflammatory leukocytes, which can increase inflammation responses and tissue damage. In addition, ROS (such as O_2_^·−^, H_2_O_2_, and HO·) secreted by phagocytes can accumulate at the sites of inflammation, leading to lipid peroxidation [49]. Moreover, the unique chemical structure of QUE has rendered its antioxidant activity one of the most extensively studied biochemical features of QUE [58], prompting further investigation into its effects on inhibiting oxidative stress induced by DSS in the intestine. In the present study, we observed a significant increase in MDA levels in the colon of DSS-treated mice compared to the mice in the control group, indicating extensive ROS production in the colonic tissues. To mitigate ROS-induced tissue damage, the body has endogenous antioxidant defense mechanisms, such as antioxidant enzymes including SOD, GSH, and CAT, which can protect against oxidative damage to tissues caused by ROS [59]. Several studies have demonstrated that the relationship between oxidative stress and the antioxidant system in colitis is very close [7,60]. Research conducted by Chen et al. [6] and Arda-Pirincci et al. [60] revealed that the MDA level in the colon was elevated in cases of colitis, while the activities of SOD, CAT, and GSH-Px were decreased in the colitis group compared to controls. Our findings showed that the level of MDA in the colon significantly increased after DSS treatment, while antioxidant levels markedly decreased, which was consistent with previous reports [7]. Meanwhile, we also found that QUE can reduce the level of MDA and increase the levels of CAT, SOD, and GSH to alleviate colon damage induced by oxidative stress. Nrf2, a key modulator of oxidative stress, translocates to the nucleus and binds to specific DNA sequences to regulate the transcription of downstream antioxidant enzymes, such as HO-1 and NAD(P)H quinone dehydrogenase 1 (NQO1), thus protecting cells from oxidative damage [34]. Usually, Nrf2 is stored in the cytoplasm and combines with the inhibitory protein Kelch-like ECH-associated protein 1 (Keap1) in a normal state. Once stimulated, Nrf2 dissociates from Keap1 and translocates from the cytoplasm to the nucleus, where it binds to the antioxidant response element (ARE) to induce HO-1 transcription [61]. As a cytoprotective and anti-inflammatory enzyme, the level of HO-1 level is elevated during inflammatory responses, and the upregulation of HO-1 can suppress the M1 phenotype molecules macrophage overproduction [62], which aligns with our findings. It has been reported that the Nrf2/HO-1 pathway exhibits a negative correlation with the expression of pro-inflammatory cytokines such as IL-1β, IL-6, and TNF-α in microglia, and the upregulation of Nrf2/HO-1 has been associated with improvements in depressive-like behavior [63]. Our data also indicate that the upregulation of Nrf2/HO-1 corresponds with a decrease in inflammatory cytokine levels. In summary, our findings illustrate that QUE effectively increases Nrf2 and HO-1 expression levels, as well as antioxidant activity in DSS-induced mice, which may elucidate its protective effects against colitis.

There is increasing evidence that the gut microbiota and the host have a symbiotic relationship in healthy individuals, while gut microbiota dysbiosis is implicated in the pathogenesis of IBD [64]. In this context, we performed a 16S rRNA analysis, which revealed that the microbial community composition and diversity within the colon were significantly changed following DSS and QUE administration. PCoA analysis showed that the gut microbiota in the DSS-induced mice group exhibited dramatic changes compared to the control group, but QUE administration mitigated the gut microbiota shifts induced by the DSS challenge. Among these notable changed bacteria, *Bacteroidaceae* and *Bacteroides* were predominant in the DSS group, but were normalized by QUE supplementation. Previous studies have reported an increase in the proportion of the *Bacteroidaceae* and *Bacteroides* in both patients and experimental animal models of IBD [29,65]. *Bacteroides* species are recognized for their role in infections across various anatomical sites and have been isolated from numerous patients suffering from meningitis and brain abscesses. Following their entry into the bloodstream during extraintestinal infections, these microbes may penetrate the central nervous system (CNS) by traversing the blood–brain barrier via the olfactory and trigeminal cranial nerves. Additionally, they have also been linked to oral infections and abscesses in the cervical region. In approximately 90% of lung abscess cases, polymicrobial infections are observed, with *Bacteroides fragilis* identified as the predominant anaerobe isolated [66]. Furthermore, *Bacteroides vulgatus* and *Bacteroides fragilis* have been identified as the two primary isolates from patients with Crohn’s disease, with the latter being associated with intra-abdominal abscesses, appendicitis, and IBD [66]. In instances of compromised immune systems, disruption of the intestinal barrier, or excessive antibiotic use, *Bacteroidaceae* can translocate through the intestinal mucosa into normally sterile tissues, potentially leading to various disease conditions with an associated mortality rate exceeding 19% [67]. In our study, we observed a negative correlation between the abundance of *Bacteroidaceae* and *Bacteroides* and the levels of TJ proteins and IL-10, while a positive correlation was noted with inflammatory markers of colitis. The findings indicate that the protective effect of QUE on the repair of intestinal damage induced by DSS may largely depend on the regulation of *Bacteroidaceae* and *Bacteroides*, although further research is warranted to ascertain which specific microbiota is more dominant. Additionally, we found a significant positive correlation between the abundance of *Akkermansia* and the level of mucin 2 (MUC2) in the colon. *Akkermansia* has been identified as a potential probiotic agent that improves mucosal inflammation by improving DSS-induced gut barrier function and microbial community composition in rats [68]. The MUC2 mucus barrier serves as the first barrier to prevent direct contact between intestinal bacteria and colonic epithelial cells [69]. Therefore, we speculate that the protective and healing effects of QUE following mucosal damage may be mediated through improving *Akkermansia* abundance in the intestine, and this result still needs to be studied further. Most commonly, *Odoribacter* is known for its ability to improve the intestinal barrier and relieve colitis by producing butyrate [70,71,72]. Chen et al. also reported a positive correlation between *Odoribacter* and the level of occludin, while a negative correlation was observed between *Odoribacter* and the level of IL-6 [7], which is similar to our studies. It has been established that small molecule metabolites derived from bacterial communities within the gastrointestinal tract serve as critical intermediates that facilitate interactions between gut microbiota and host organisms. Short-chain fatty acids, bile acids, and tryptophan catabolites, which are regulated by gut microbiota, are significantly impacted during the progression of IBD [68]. In this study, we conducted an analysis of the potential metabolic pathways influenced by QUE, utilizing the KEGG and MetaCyc databases. Our findings indicate that QUE primarily regulates pathways associated with amino acid metabolism and biosynthesis, carbohydrate metabolism and biosynthesis, cofactor and vitamin metabolism and biosynthesis, as well as nucleotide metabolism and biosynthesis in microbial systems. Furthermore, future research will focus on elucidating the mechanisms through which gut microbiota mediate the effects of QUE in mitigating colitis by regulating these metabolites.

## 5. Conclusions

The current results collectively demonstrate that the primary mechanisms through which QUE significantly alleviated DSS-induced clinical symptoms and colon injury include the inhibition of colonic inflammation, the preservation of intestinal barrier integrity, and the regulation of oxidative stress and gut microbiota (Figure 9). This study elucidates the relationship between intestinal microbiota and environmental factors that influence intestinal integrity, including oxidative stress and the inflammatory mucosal barrier, through association analysis. Furthermore, it investigates the mechanisms by which QUE mitigates intestinal injury, thereby offering a theoretical foundation for the clinical application of QUE in the treatment of colitis.

## Figures and Tables

**Figure 1 microorganisms-12-01973-f001:**
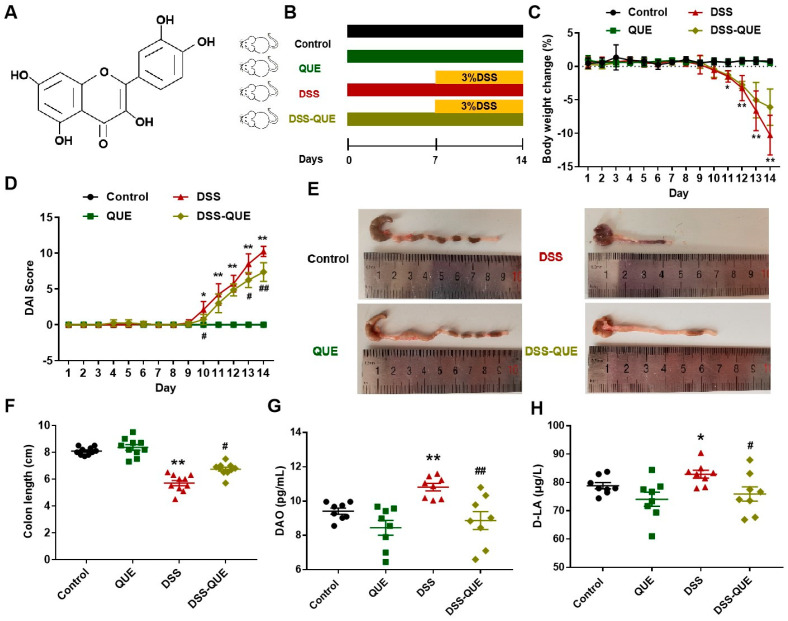
QUE treatment relieved DSS-induced colitis. (**A**) QUE chemical structure. (**B**) Experiment treatment timeline. (**C**) The change in body weight. (**D**) Disease activity index (DAI) score of mice. (**E**) Representative morphological images of colon tissue. (**F**) The length of the colon. (**G**,**H**) Serum levels of DAO and D-LA. The results are expressed as the mean ± SEM (n = 8). * *p* < 0.05, ** *p* < 0.01 vs. the Control group; # *p* < 0.05, ## *p* < 0.01 vs. the DSS group.

**Figure 2 microorganisms-12-01973-f002:**
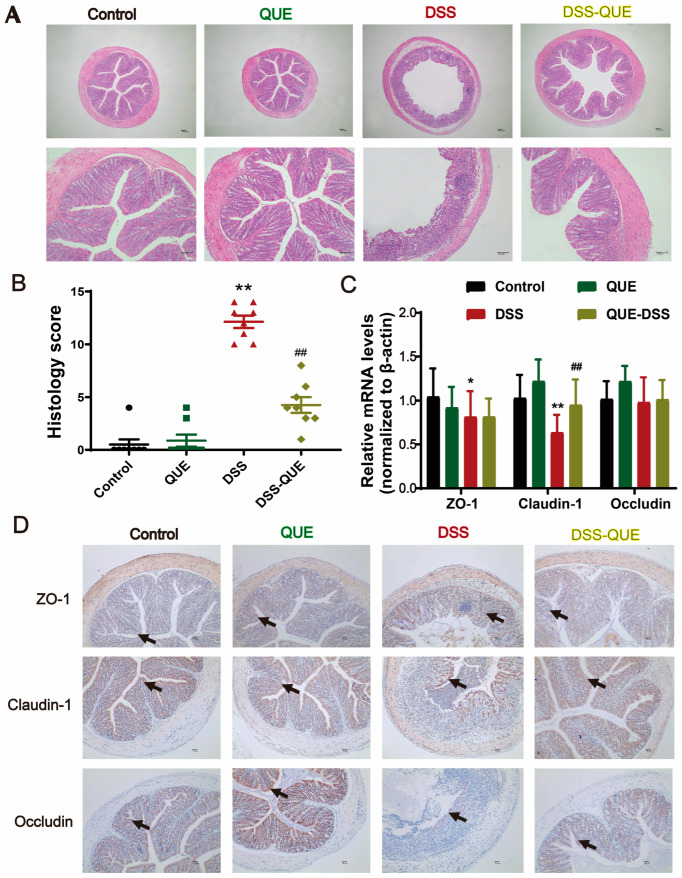
Effect of QUE on TJ proteins in the colon. (**A**) H&E staining of colon tissue sections from each group, original magnification, above: ×40, below: ×100. (**B**) Histological scoring of the colon (n = 8). (**C**) Relative mRNA expression of ZO-1, Claudin-1, and Occludin in the colon of mice was detected by quantitative real-time PCR (n = 4). (**D**) Immunohistochemistry analysis on ZO-1, Claudin-1, and Occludin in colon sections from different groups (n = 3), original magnification ×100. * *p* < 0.05, ** *p* < 0.01 vs. the control group; ## *p* < 0.01 vs. the DSS group. The arrow represents where the protein is expressed.

**Figure 3 microorganisms-12-01973-f003:**
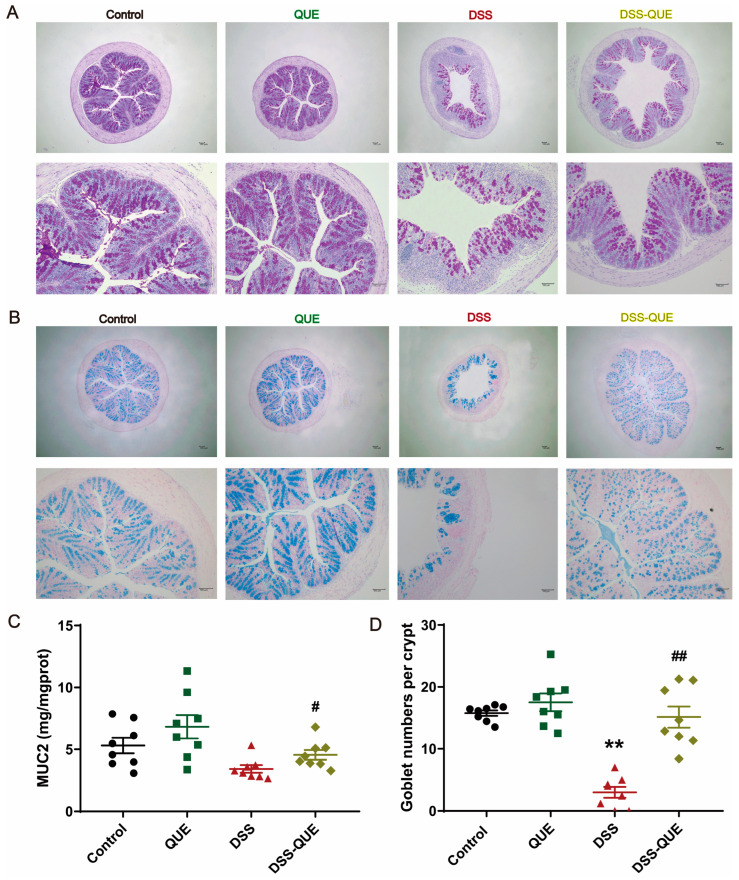
Effects of QUE on the mucous layer. (**A**) Alcian blue staining, original magnification, above: ×40, below: ×100. (**B**) Histological sections of the colon (stained with PAS), original magnification, above: ×40, below: ×100. (**C**) Concentration of MUC2. (**D**) Number of goblet cells. The results are expressed as the mean ± SEM (n = 8). ** *p* < 0.01 vs. the control group; # *p* < 0.05, ## *p* < 0.01 vs. the DSS group.

**Figure 4 microorganisms-12-01973-f004:**
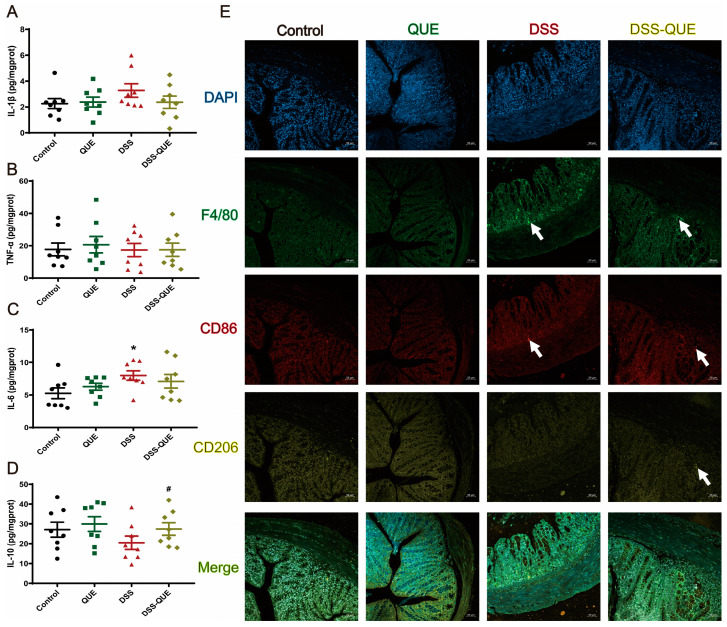
QUE regulated inflammation and macrophage polarization in DSS-induced colitis mice. (**A**) IL-1β level in colon. (**B**) IL-6 level in the colon. (**C**) TNF-α level in the colon. (**D**) IL-10 level in the colon. The results are expressed as the mean ± SEM (n = 8). (**E**) Immunostained with DAPI (blue), F4/80 (green), CD86 (red), and CD206 (yellow) in colon tissue, original magnification ×100 (n = 3). * *p* < 0.05 vs. the control group; # *p* < 0.05 vs. the DSS group. The arrow represents where the protein is expressed.

**Figure 5 microorganisms-12-01973-f005:**
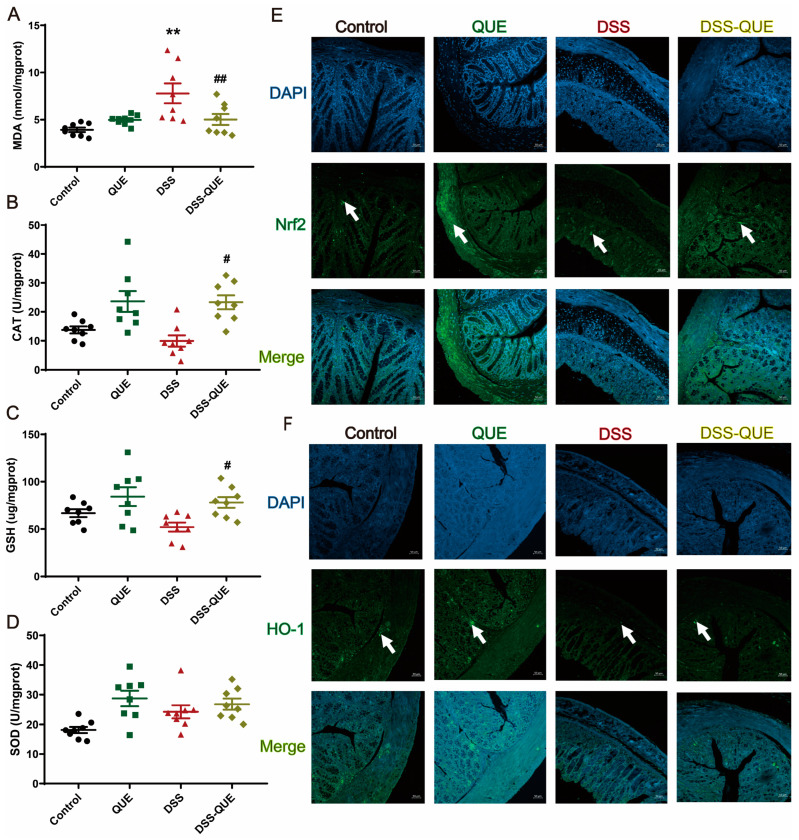
Effects of QUE on the activity of oxidative stress-related enzymes and the Nrf2/HO-1 pathway in the colon. (**A**) The level of MDA in the colon. (**B**) The activity of CAT in the colon. (**C**) The level of GSH in the colon. (**D**) The activity of SOD in the colon. The results are expressed as the mean ± SEM (n = 8). (**E**) Representative immunofluorescence images of Nrf2 in the colon (n = 3), original magnification ×100. (**F**) Representative immunofluorescence images of HO-1 in the colon (n = 3), original magnification ×100. ** *p* < 0.01 vs. the control group; # *p* < 0.05, ## *p* < 0.01 vs. the DSS group. The arrow represents where the protein is expressed.

**Figure 6 microorganisms-12-01973-f006:**
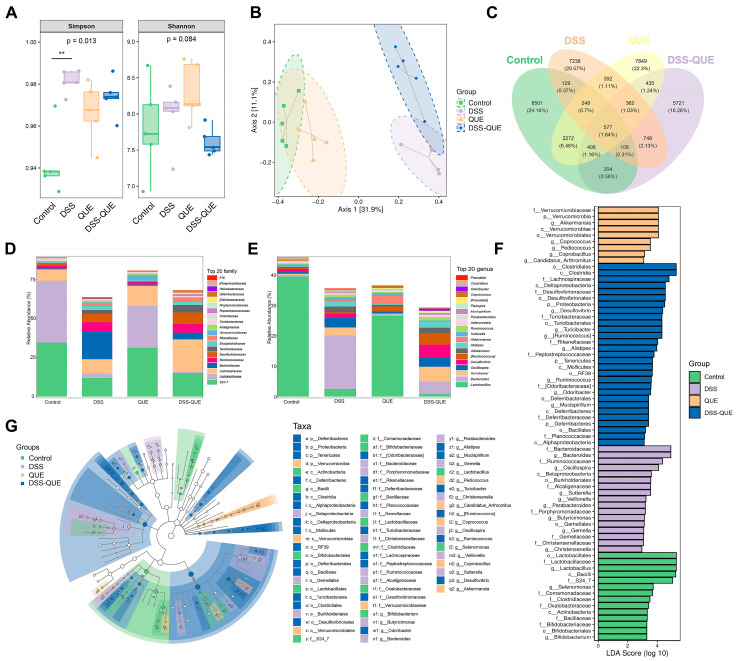
The composition of the colon microbiota was altered by QUE administration in colitis mice. (**A**) Shannon and Simpson indexes were used to estimate alpha diversity. (**B**) Gut microbiota principal component analysis (PCA) plot. (**C**) Venn diagram of ASVs. Relative abundance of predominant bacteria was shown at the family (**D**) and genus (**E**) levels. (**F**) Taxonomic cladogram of LEfSe analysis. (**G**) Linear discriminant analysis (LDA) score for different taxa abundances. Data were expressed as means ± SEM (n = 5). ** *p* < 0.01.

**Figure 7 microorganisms-12-01973-f007:**
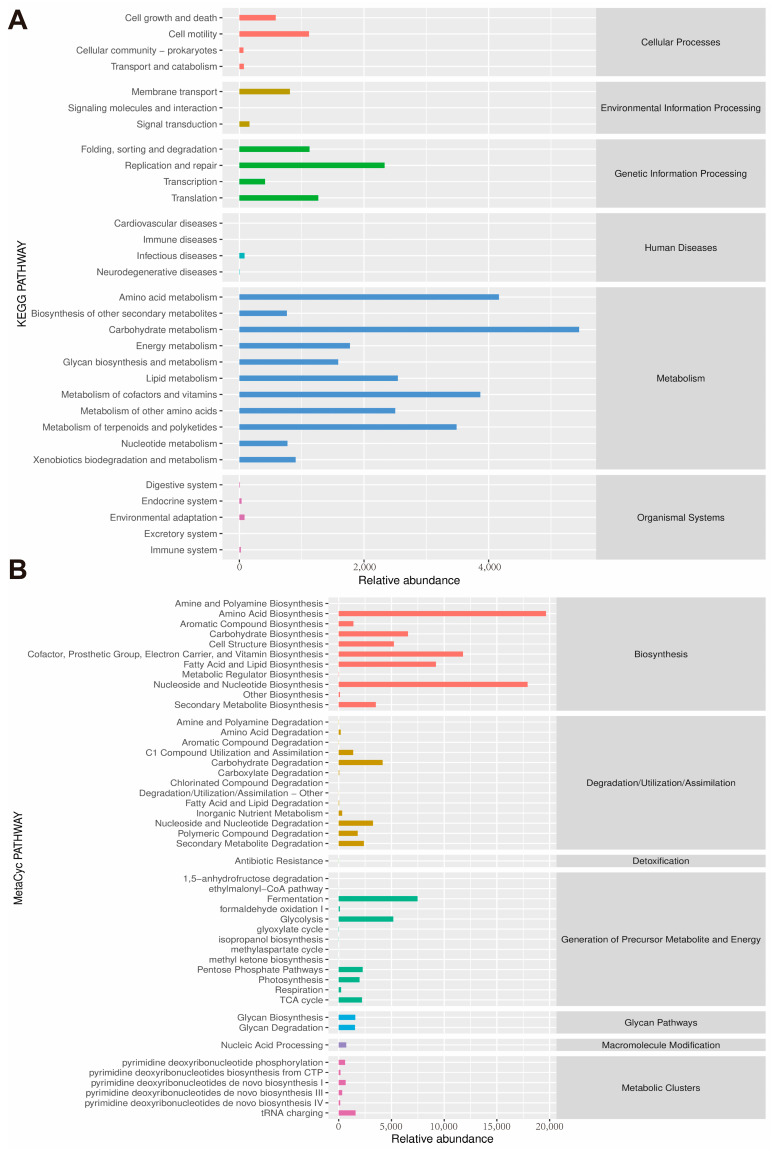
The pathway of different abundances of microbiota. (**A**) The pathway of different abundances of microflora at KEGG levels. (**B**) The pathway of different abundances of microflora at MetaCyc levels.

**Figure 8 microorganisms-12-01973-f008:**
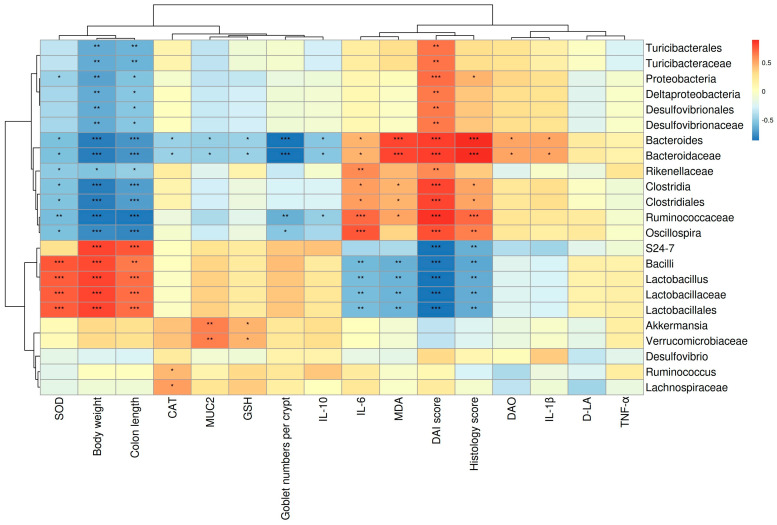
Correlation analysis between the 23 most dominant genera and micro-environmental factors. The positive and negative correlations were represented by the red and blue blocks, respectively. n = 5 for each group. * *p* < 0.05, ** *p* < 0.01, *** *p* < 0.001.

**Figure 9 microorganisms-12-01973-f009:**
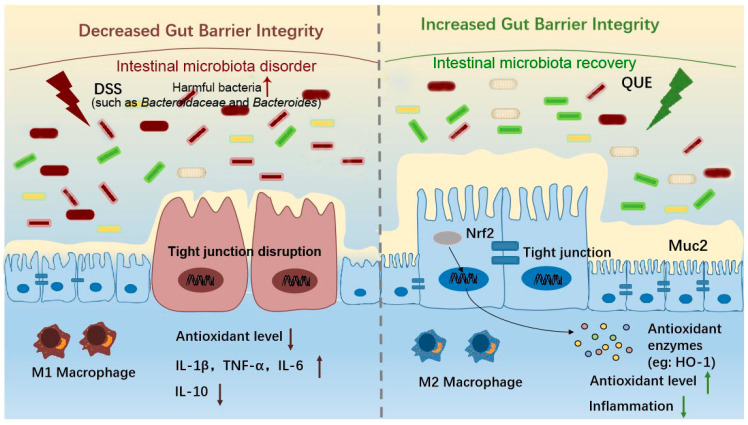
Possible mechanism of QUE to treat colitis in mice.

## Data Availability

The datasets generated for this study can be found in online repositories. The names of the repository/repositories and accession number(s) can be found at: https://www.ncbi.nlm.nih.gov/genbank/, PRJNA867489, accessed on 10 August 2022.

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
