# Peer review of "Network Analysis of Gut Microbial Communities Reveals Key Reason for Quercetin Protects against Colitis"

_microorganisms, 2024, doi:10.3390/microorganisms12101973_

Round 1
Reviewer 1 Report
Comments and Suggestions for Authors
The manuscript titled "Network Analysis of Gut Microbial Communities Reveal Key..." by Yanan Lv et al. is quite a good work. However, I believe that it needs a fairly comprehensive revision before it can be published.
Below are my comments, suggestions, and questions for the authors:
- The title is too long; it should be revised to make it more attractive.
- The first sentence of the abstract could be written better, much better, in fact.
- In general, the abstract could be rewritten to be more polished. As it stands, it is somewhat chaotic.
- Is "Inflammatory bowel disease" really necessary as a keyword?
- The introduction could be slightly expanded. There is no shortage of data online about Quercetin and its derivatives; I believe the authors would agree with me on this.
- The last paragraph of the introduction should be properly revised to include specific objectives of the study, rather than being presented superficially and half-heartedly.
- Figure 1 – The structure of Quercetin is poorly drawn, of low quality. This needs to be corrected. In general, the entire figure is of poor quality.
- Figure 6 – Same issue, very poor quality of the figure and everything is so small. It's hard to see anything. Please improve it.
- Figure 7 – Same as Figure 6.
- The discussion is very well written, and there's no need to critique it.
- The conclusion definitely needs improvement, as it currently doesn't convey much. Please put more effort into it.
In summary, I believe the paper is valuable but has several shortcomings that need to be addressed as soon as possible before the manuscript can proceed to publication. Some parts of the work were approached rather poorly by the authors.
Author Response
Dear Reviewer:
We are very grateful for your efforts on our manuscript and for giving us the opportunity to resubmit a revised version of our manuscript. These comments are all valuable and very helpful for revising and improving our manuscript, as well as the important guiding significance to our researches. According with your advice, we amended the relevant part in manuscript. Some of your questions were answered below.
We hope that our manuscript is now suitable for publication in Microorganisms.
Manuscript ID number:
3160059
Title of paper:
Network Analysis of Gut Microbial Communities Reveal Key Reason for Quercetin Protects Against Colitis
- The title is too long; it should be revised to make it more attractive.
Response: Thank you for your suggestions. We have We have changed the original title “Network Analysis of Gut Microbial Communities Reveal Key Reason for Quercetin Protects Against Dextran Sodium Sulfate (DSS)-induced Colitis Mice” to the current title “Network Analysis of Gut Microbial Communities Reveal Key Reason for Quercetin Protects Against Colitis ” in the subsequently submitted manuscripts.
- The first sentence of the abstract could be written better, much better, in fact.
Response: Thank you for your suggestions. We have been modified the first sentence of the abstract, you can observe it in line 18-19 of the subsequently submitted manuscripts.
- In general, the abstract could be rewritten to be more polished. As it stands, it is somewhat chaotic.
Response: Thank you for your suggestions. We have amended the abstract in the later submitted manuscripts.
- Is "Inflammatory bowel disease" really necessary as a keyword?
Response: Thank you for your suggestions. We have replaced “Inflammatory bowel disease” with “colitis” in the keywords of the subsequently submitted manuscripts.
- The introduction could be slightly expanded. There is no shortage of data online about Quercetin and its derivatives; I believe the authors would agree with me on this.
Response: Thank you for your suggestions. We have expanded the introduction and added studies on quercetin's pharmacological effects in the subsequently submitted manuscripts.
- The last paragraph of the introduction should be properly revised to include specific objectives of the study, rather than being presented superficially and half-heartedly.
Response: Thank you for your suggestions. We have rewrote the last paragraph of the introduction in the subsequently submitted manuscripts.
- Figure 1 – The structure of Quercetin is poorly drawn, of low quality. This needs to be corrected. In general, the entire figure is of poor quality.
Response: Thank you for your suggestions. We have made several modifications to enhance the clarity of Figure 1 and have resubmitted the revised image accordingly.
- Figure 6 – Same issue, very poor quality of the figure and everything is so small. It's hard to see anything. Please improve it.
Response: Thank you for your suggestions. We have made several modifications to enhance the clarity of Figure 6 and have resubmitted the revised image accordingly. Furthermore, due to the substantial amount of content presented in Figure 6, the font size within the image has been minimized. Therefore, I kindly request that you enlarge the image for better visibility. I appreciate your understanding regarding any inconvenience this may cause.
- Figure 7 – Same as Figure 6.
Response: Thank you for your suggestions. We have made several modifications to enhance the clarity of Figure 7 and have resubmitted the revised image accordingly.
- The discussion is very well written, and there's no need to critique it.
Response: Thank you for your suggestions.
- The conclusion definitely needs improvement, as it currently doesn't convey much. Please put more effort into it.
Response: Thank you for your suggestions. We have rewritten the conclusion in the subsequently submitted manuscripts..
In summary, I believe the paper is valuable but has several shortcomings that need to be addressed as soon as possible before the manuscript can proceed to publication. Some parts of the work were approached rather poorly by the authors.
Response: Thank you for your suggestions for our manuscripts. We have undertaken extensive revisions and enhancements, and the modified sections are highlighted in red font in the subsequently submitted manuscripts.

Reviewer 2 Report
Comments and Suggestions for Authors
There are many English language errors in both the abstract and the full-text, which need careful revision and correction.
Lines 22-23: “…QUE oral gavage administration ameliorates the symptoms and histopathological change of colitis…” . This sounds weird, it should be something along the line of “QUE oral gavage administration ameliorates the symptoms and the histopathological changes associated with colitis….”.
Lines 21 and 29: “DSS-induced mice” should be “DSS-induced colitis mice”.
Line 40: “the symptom of IBD patients present with abdominal pain…” needs also correction (e.g. IBD patient present with symptoms such as abdominal pain…”.
We will not review each sentence, but there are many such examples that need revision and correction.
One of the biggest issues with this paper is that the introduction does not cite the relevant scientific literature. Relatively many scientific papers (we’ve counted at least nine) have described the effect of QUE on colitis in mice and have explored its mechanisms of activity, including transcriptomics and gut microbiota diversity. In this context, it is the authors’ responsibility to review the previous research and clarify what their own research brings new. The discussion section also does not include such discussions of the available literature on quercetin experiments with DSS-induced colitis in mice.
Line 417: the authors should clarify how they have selected the 50 mg/kg dose.
Line 425: “Mice were weighed daily until sacrificed”. At what day were the animals sacrificed? Was there any anesthesia used in the killing process?
Line 439-442: While the authors have followed the procedure recommended by the manufacturer, they should at least clarify what equipment they used during the procedure. The same holds true for lines 443-447 and 448-451.
Lines 506-509: if the authors have used one-way ANOVA for their statistical analysis, they should also report on the post-hoc tests used. Also, labeling p values lower than 0.01 as “extremely significant” is rather extreme; “highly significant” should be sufficient.
Lines 107-108: the authors claim results based on diamine oxidase (DAO) and D-lactic acid (D-LA) measurements, but their methods do not describe any such measurements.
Line 126: please clarify the use of abbreviations with their first use, in this case TJ. The same for ASVs (line 208), LDA (line 211),
In Figure 2, we hope to be wrong, but the figure from the QUE group corresponding to the ZO-1 and the picture from the Control group corresponding Occluding seem to be one and the same image (although they are attributed to animals from different treatment groups).
Figure 6: the fonts used are too small to allow understanding (and zooming in they become blurred).
Comments on the Quality of English Language
Extensive English language review is necessary, almost every sentence needs at least some correction, whereas some sentences are rather difficult to understand.
Author Response
Dear reviewer:
We are very grateful for your efforts on our manuscript and for giving us the opportunity to resubmit a revised version of our manuscript. These comments are all valuable and very helpful for revising and improving our manuscript, as well as the important guiding significance to our researches. According with your advice, we amended the relevant part in manuscript. Some of your questions were answered below.
We hope that our manuscript is now suitable for publication in Microorganisms.
Manuscript ID number:
3160059
Title of paper:
Network Analysis of Gut Microbial Communities Reveal Key Reason for Quercetin Protects Against Colitis
- There are many English language errors in both the abstract and the full-text, which need careful revision and correction.
Response: Thank you for your valuable suggestions for our manuscripts. We have undertaken extensive revisions and enhancements, and the modified sections are highlighted in red font in the subsequently submitted manuscripts.
- Lines 22-23: “…QUE oral gavage administration ameliorates the symptoms and histopathological change of colitis…” . This sounds weird, it should be something along the line of “QUE oral gavage administration ameliorates the symptoms and the histopathological changes associated with colitis….”.
Response: Thank you for your suggestions. We have implemented the corrections, which can be observed on lines 21-22 of the subsequently submitted manuscript.
- Lines 21 and 29: “DSS-induced mice” should be “DSS-induced colitis mice”.
Response: Thank you for your suggestions. We have implemented the corrections, which can be observed on lines 20 and 29 of the subsequently submitted manuscript.
- Line 40: “the symptom of IBD patients present with abdominal pain…” needs also correction (e.g. IBD patient present with symptoms such as abdominal pain…”.
Response: Thank you for your suggestions. We have implemented the corrections, which can be observed on lines 40 of the subsequently submitted manuscript.
- We will not review each sentence, but there are many such examples that need revision and correction.
Response: Thank you for your valuable suggestions for our manuscripts. We have undertaken extensive revisions and enhancements, and the modified sections are highlighted in red font in the subsequently submitted manuscripts.
- One of the biggest issues with this paper is that the introduction does not cite the relevant scientific literature. Relatively many scientific papers (we’ve counted at least nine) have described the effect of QUE on colitis in mice and have explored its mechanisms of activity, including transcriptomics and gut microbiota diversity. In this context, it is the authors’ responsibility to review the previous research and clarify what their own research brings new. The discussion section also does not include such discussions of the available literature on quercetin experiments with DSS-induced colitis in mice.
Response: Thank you for your suggestions. We have made enhancements to the introduction and discussion sections, as well as incorporated new references, which are highlighted in red in the subsequently submitted manuscripts.
- Line 417: the authors should clarify how they have selected the 50 mg/kg dose.
Response: Thank you for your suggestions. Considering that colitis is a chronic disease, we selected the dosage based on the findings from literature “Quercetin attenuates the activation of hepatic stellate cells and liver fibrosis in mice through modulation of HMGB1-TLR2/4-NF-κB signaling pathways” and “Oral administration of quercetin inhibits bone loss in rat model of diabetic osteopenia”, which also examine chronic diseases. Additionally, we administered 100 mg/kg of quercetin via gavage to the mice. However, upon evaluating the mental state, colon length, and fecal consistency of the mice, we determined that the effects of this dosage were suboptimal. Hematoxylin and eosin (HE) staining corroborated our experimental results. We concluded that a dosage of 100 mg/kg of quercetin is excessive and does not yield favorable outcomes in the treatment of DSS-induced colitis. Consequently, we opted to utilize a dosage of 50 mg/kg of quercetin for the subsequent studies.
- Line 425: “Mice were weighed daily until sacrificed”. At what day were the animals sacrificed? Was there any anesthesia used in the killing process?
Response: Thank you for your suggestions. The mice were euthanized on the morning of the fifteenth day, following a total experimental duration of fourteen days. Prior to euthanasia, the mice were administered 3.5% chloral hydrate at a dosage of 0.1 mL/10 g of body weight.
- Line 439-442: While the authors have followed the procedure recommended by the manufacturer, they should at least clarify what equipment they used during the procedure. The same holds true for lines 443-447 and 448-451.
Response: Thank you for your suggestions. We have supplemented the instruments utilized in the experiment.
- Lines 506-509: if the authors have used one-way ANOVA for their statistical analysis, they should also report on the post-hoc tests used. Also, labeling p values lower than 0.01 as “extremely significant” is rather extreme; “highly significant” should be sufficient.
Response: Thank you for your suggestions. This part has been corrected, and it can be observed in lines 622 to 623.
- Lines 107-108: the authors claim results based on diamine oxidase (DAO) and D-lactic acid (D-LA) measurements, but their methods do not describe any such measurements.
Response: Thank you for your suggestions. We have supplemented the method how to detect the levels of DAO and D-LA, as well as the instruments utilized in the experiment, and it can be observed in lines 533 to 537 of the subsequently submitted manuscripts.
- Line 126: please clarify the use of abbreviations with their first use, in this case TJ. The same for ASVs (line 208), LDA (line 211),
- Response: Thank you for your suggestions. This mistakehas been corrected, and it can be observed in lines 249 and 255 of the subsequently submitted manuscripts.
- In Figure 2, we hope to be wrong, but the figure from the QUE group corresponding to the ZO-1 and the picture from the Control group corresponding Occluding seem to be one and the same image (although they are attributed to animals from different treatment groups).
Response: Thank you for your suggestions.We have made modifications to the image and have resubmitted it.
- Figure 6: the fonts used are too small to allow understanding (and zooming in they become blurred).
Response: Thank you for your suggestions. We have made several modifications to enhance the clarity of Figure 6 and have resubmitted the revised image accordingly. Furthermore, due to the substantial amount of content presented in Figure 6, the font size within the image has been minimized. Therefore, I kindly request that you enlarge the image for better visibility. I appreciate your understanding regarding any inconvenience this may cause.

Reviewer 3 Report
Comments and Suggestions for Authors
Dear author, the information presented is relevant, however, I consider it appropriate to include additional information to complement your article:
Introduction section
- Line 44 indicates a high prevalence of IBD, it is considered appropriate to include worldwide statistical data.
- Line 53 indicates the use of probiotics for the treatment of this pathology, it is considered appropriate to include an example of them.
-In line 80 it is described that quercetin (QU) promotes the health of the intestinal microbiota, however, the mechanism is not clear, could you indicate studies that describe it, in the same sense, in line 84, it is described that QU regulates the mycobial structure and metabolism, could you indicate to which microorganisms and which metabolic pathways are referred?
Methodology section
It is necessary to indicate throughout the section the brand name of the chemical reagents used, for example, DSS, among others.
4.1. Animals and Experimental Design
-Can you indicate why this dose of quercetin 50 mg/kg BW was considered, and include what was the vehicle for it?
-It is considered appropriate to substitute the word sacrifice for euthanasia and to indicate how the organs were stored.
4.3. Level of Cytokines in Colon Tissues
-It is considered adequate to indicate how the protein was obtained from the organs, what type of methodology was used, if protease inhibitors were included, at what temperature the collection was performed and if it was total, membrane or soluble protein that was analyzed.
- And in this same sense, what equipment was used for its quantification and other specifications such as the reading Abs, etc,
4.4. Level of Mucin2 (MUC2) in Colon Tissues
-In this section it is also considered appropriate to indicate if the sample was the same protein used, and the parameters of measurement of the MUC2 as well as the equipment used.
4.5. Antioxidant Enzyme Activity Determination
-It is necessary to include more information about the methodology, although it is described that it was performed with a commercial kit, it is important to include information about how the protein was obtained for this determination, was it the same as for the other markers, or did it have another process?
4.6. Alcian Blue and Periodic Acid-schiff (PAS) Staining
-Could you include more information on the technique, as well as the type of microscope used to observe the histological samples?
4.8. Immunohistochemistry
-Is it necessary to include the brand of microscope used?
4.10. Gut Microbiota Analysis.
-It is necessary to include information on the type of sequencer used.
Results section
2.1. QUE Treatment Improved the Symptoms of DSS-induced Colitis
-I believe it is necessary to contextualize why the chemical structure of QUE (line 94) is considered relevant as part of the results.
2.2 QUE Alleviated Colonic Tissues Damage Caused by DSS and Protected the Intestinal Barrier
-It is not clear how the Histological scoring of colon (n=8) was obtained (Figure 2B), could you indicate how this result was generated?
2.5. Regulation of Gut Microbiota Composition by QUE
-It is appropriate to change the word bacteria instead of microbes (219).
-It is suggested to change intestinal microbiota instead of microflora in the caption of figure 7 (line 235).
-It is important to revise the document to modify the writing of the names of bacteria to italics, especially when including the names of species such as Akkermansia, among others.
-In lines 248-251 it is described the importance that it has“, we also found that the abundance of Akkermansia has a significant positive correlation with the MUC2 level of colon, which indicated that Akkermansia maybe a benefit bacteria for protective and healing effects after mucosal damage” it is necessary to include the bibliographic citation that describes this effect.
- Figure 6 contains very relevant information, however, it is not possible to adequately observe the data, hopefully it could have a higher resolution.
Discussion
- Could include more information on the side effects generated by the consumption of IBD treatments described in lines (264-264).
-Could indicate more information on the therapeutic effect of WHAT, such as its mechanism of action, related to its chemical structure, as described by various authors.
-I consider it relevant to indicate why the concentration of quercetin was used in the model and, based on this, discuss the results obtained.
- I consider it appropriate to include information on the mechanism of action of DSS and to establish why this concentration was used (lines 290-293).
- As part of the results, the KEGG analysis and Meta-Cyc analysis is described, indicating the impact on the biosynthesis of metabolites of different metabolic pathways, however these results are not discussed, I consider it appropriate to include this information.
- In lines 388-390 it is described that Bacteroides are related to intestinal damage and inflammation, could you indicate the possible mechanisms based on other studies?
-Also no information is discussed about the effect on pathogenic bacteria that could be modified causing intestinal dysbiosis therefore modification of the intestinal barrier, or the generation of short chain fatty acids that could be generated that are related to the integrity of the intestinal mucosa, only butyrate is included, will it be the only one that has been described?
I believe that the conclusion should be positioned at the end of the discussion and that the proposed image should be part of the discussion. In this way a concrete conclusion of the results will be generated.
Author Response
Dear reviewer:
We are very grateful for your efforts on our manuscript and for giving us the opportunity to resubmit a revised version of our manuscript. These comments are all valuable and very helpful for revising and improving our manuscript, as well as the important guiding significance to our researches. According with your advice, we amended the relevant part in manuscript. Some of your questions were answered below.
We hope that our manuscript is now suitable for publication in Microorganisms.
Manuscript ID number:
3160059
Title of paper:
Network Analysis of Gut Microbial Communities Reveal Key Reason for Quercetin Protects Against Colitis
- Introduction section
- Line 44 indicates a high prevalence of IBD, it is considered appropriate to include worldwide statistical data.
Response: Thank you for your suggestions. I have already supplemented this part in the introduction, and it can be observed in lines 45 to 49 of the subsequently submitted manuscripts.
- Line 53 indicates the use of probiotics for the treatment of this pathology, it is considered appropriate to include an example of them.
Response: Thank you for your suggestions. I have already supplemented this part in the introduction, and it can be observed in lines 58 to 59 of the subsequently submitted manuscripts.
-In line 80 it is described that quercetin (QU) promotes the health of the intestinal microbiota, however, the mechanism is not clear, could you indicate studies that describe it, in the same sense,
Response: Thank you for your suggestions. I have already supplemented this part in the introduction, and it can be observed in lines 92 to 95 of the subsequently submitted manuscripts.
in line 84, it is described that QU regulates the mycobial structure and metabolism, could you indicate to which microorganisms and which metabolic pathways are referred?
Response: Thank you for your suggestions. I have already supplemented this part in the introduction, and it can be observed in lines 100 to 104 of the subsequently submitted manuscripts.
Methodology section
It is necessary to indicate throughout the section the brand name of the chemical reagents used, for example, DSS, among others.
Response: Thank you for your suggestions. I have already added the brand name of the chemical reagents we used in the subsequently submitted manuscripts.
4.1. Animals and Experimental Design
-Can you indicate why this dose of quercetin 50 mg/kg BW was considered, and include what was the vehicle for it?
Response: Thank you for your suggestions. Considering that colitis is a chronic disease, we selected the dosage based on the findings from literature and B, which also examine chronic diseases. Additionally, we administered 100 mg/kg of quercetin via gavage to the mice. However, upon evaluating the mental state, colon length, and fecal consistency of the mice, we determined that the effects of this dosage were suboptimal. Hematoxylin and eosin (HE) staining corroborated our experimental results. We concluded that a dosage of 100 mg/kg of quercetin is excessive and does not yield favorable outcomes in the treatment of DSS-induced colitis. Consequently, we opted to utilize a dosage of 50 mg/kg of quercetin for the subsequent studies.
-It is considered appropriate to substitute the word sacrifice for euthanasia and to indicate how the organs were stored.
Response: Thank you for your suggestions. I have already supplemented this part in the introduction, and it can be observed in lines 512 to 519 of the subsequently submitted manuscripts.
4.3. Level of Cytokines in Colon Tissues
-It is considered adequate to indicate how the protein was obtained from the organs, what type of methodology was used, if protease inhibitors were included, at what temperature the collection was performed and if it was total, membrane or soluble protein that was analyzed.
Response: Thank you for your suggestions. I have already supplemented this part in the introduction, and it can be observed in materials and methods of the subsequently submitted manuscripts.
- And in this same sense, what equipment was used for its quantification and other specifications such as the reading Abs, etc,
Response: Thank you for your suggestions. We have supplemented the instruments utilized in this experiment.
4.4. Level of Mucin2 (MUC2) in Colon Tissues
-In this section it is also considered appropriate to indicate if the sample was the same protein used, and the parameters of measurement of the MUC2 as well as the equipment used.
Response: Thank you for your suggestions. I have already supplemented this part in the introduction, and it can be observed in lines 548 to 551 of the subsequently submitted manuscripts.
4.5. Antioxidant Enzyme Activity Determination
-It is necessary to include more information about the methodology, although it is described that it was performed with a commercial kit, it is important to include information about how the protein was obtained for this determination, was it the same as for the other markers, or did it have another process?
Response: Thank you for your suggestions. I have already supplemented this part in the introduction, and it can be observed in lines 553 to 557 of the subsequently submitted manuscripts.
4.6. Alcian Blue and Periodic Acid-schiff (PAS) Staining
-Could you include more information on the technique, as well as the type of microscope used to observe the histological samples?
Response: Thank you for your suggestions. We have supplemented the instruments utilized in this experiment.
4.8. Immunohistochemistry
-Is it necessary to include the brand of microscope used?
Response: Thank you for your suggestions. We have supplemented the instruments utilized in this experiment.
4.10. Gut Microbiota Analysis.
-It is necessary to include information on the type of sequencer used.
Response: Thank you for your suggestions. We have supplemented the instruments utilized in this experiment.
Results section
2.1. QUE Treatment Improved the Symptoms of DSS-induced Colitis
-I believe it is necessary to contextualize why the chemical structure of QUE (line 94) is considered relevant as part of the results.
Response: Thank you for your suggestions. In this study, we did not conduct an analysis of the structural properties of quercetin; rather, we merely referenced it. If you believe that this inclusion is inappropriate for the subsequent manuscript, I am willing to remove it. I appreciate your understanding regarding any misunderstandings that may have arisen.
2.2 QUE Alleviated Colonic Tissues Damage Caused by DSS and Protected the Intestinal Barrier
-It is not clear how the Histological scoring of colon (n=8) was obtained (Figure 2B), could you indicate how this result was generated?
Response: Thank you for your suggestions. The scoring methodology is derived from the approach outlined in reference 69. The subsequent figure illustrates the reference standards as discussed in the existing literature.
2.5. Regulation of Gut Microbiota Composition by QUE
-It is appropriate to change the word bacteria instead of microbes (219).
Response: Thank you for your suggestions. We have corrected it, and it can be observed in line 260 of the subsequently submitted manuscripts.
-It is suggested to change intestinal microbiota instead of microflora in the caption of figure 7 (line 235).
Response: Thank you for your suggestions. We have corrected it, and it can be observed in line 280 of the subsequently submitted manuscripts.
-It is important to revise the document to modify the writing of the names of bacteria to italics, especially when including the names of species such as Akkermansia, among others.
Response: Thank you for your suggestions. We have corrected it in the subsequently submitted manuscripts.
-In lines 248-251 it is described the importance that it has“, we also found that the abundance of Akkermansia has a significant positive correlation with the MUC2 level of colon, which indicated that Akkermansia maybe a benefit bacteria for protective and healing effects after mucosal damage” it is necessary to include the bibliographic citation that describes this effect.
Response: Thank you for your suggestions. After thorough deliberation, we have determined that this sentence is not suitable for inclusion in the results section. Consequently, we have revised the sentence and engaged in a discussion regarding this aspect of the content, and it can be observed in lines 460 to 468 of the subsequently submitted manuscripts.
- Figure 6 contains very relevant information, however, it is not possible to adequately observe the data, hopefully it could have a higher resolution.
Response: Thank you for your suggestions. We have made several modifications to enhance the clarity of Figure 6 and have resubmitted the revised image accordingly. Furthermore, due to the substantial amount of content presented in Figure 6, the font size within the image has been minimized. Therefore, I kindly request that you enlarge the image for better visibility. I appreciate your understanding regarding any inconvenience this may cause.
Discussion
- Could include more information on the side effects generated by the consumption of IBD treatments described in lines (264-264).
Response: Thank you for your suggestions. I have already explained this part in the introduction, and it can be observed in lines 53 to 57 in the subsequently submitted manuscripts.
-Could indicate more information on the therapeutic effect of WHAT, such as its mechanism of action, related to its chemical structure, as described by various authors.
Response: Thank you for your suggestions. This section of the content has been enhanced and is indicated using red font in the subsequently submitted manuscripts .
-I consider it relevant to indicate why the concentration of quercetin was used in the model and, based on this, discuss the results obtained.
Response: Thank you for your suggestions. Considering that colitis is a chronic disease, we selected the dosage based on the findings from literature and B, which also examine chronic diseases. Additionally, we administered 100 mg/kg of quercetin via gavage to the mice. However, upon evaluating the mental state, colon length, and fecal consistency of the mice, we determined that the effects of this dosage were suboptimal. Hematoxylin and eosin (HE) staining corroborated our experimental results. We concluded that a dosage of 100 mg/kg of quercetin is excessive and does not yield favorable outcomes in the treatment of DSS-induced colitis. Consequently, we opted to utilize a dosage of 50 mg/kg of quercetin for the subsequent studies.
- I consider it appropriate to include information on the mechanism of action of DSS and to establish why this concentration was used (lines 290-293).
Response: Thank you for your suggestions. Using DSS to establish a colitis model is already a very typical method, and using 3% DSS concentration is the optimal concentration recommended by the manufacturer. Therefore, in this study, we used 3% DSS to establish a colitis model.
- As part of the results, the KEGG analysis and Meta-Cyc analysis is described, indicating the impact on the biosynthesis of metabolites of different metabolic pathways, however these results are not discussed, I consider it appropriate to include this information.
Thank you for your suggestions.We have added this section in the discussion and it can be observed in lines 472 to 484 of the subsequently submitted manuscripts.
- In lines 388-390 it is described that Bacteroides are related to intestinal damage and inflammation, could you indicate the possible mechanisms based on other studies?
Response: Thank you for your suggestions.We have added this section in the discussion and it can be observed in lines 441 to 458 of the subsequently submitted manuscripts .
-Also no information is discussed about the effect on pathogenic bacteria that could be modified causing intestinal dysbiosis therefore modification of the intestinal barrier, or the generation of short chain fatty acids that could be generated that are related to the integrity of the intestinal mucosa, only butyrate is included, will it be the only one that has been described?
Response: Thank you for your suggestions.We have added this section in the discussion and it can be observed in lines 472 to 477 the subsequently submitted manuscripts .
I believe that the conclusion should be positioned at the end of the discussion and that the proposed image should be part of the discussion. In this way a concrete conclusion of the results will be generated.
Response: Thank you for your suggestions.We have revised this part of the subsequently submitted manuscripts .

Round 2
Reviewer 1 Report
Comments and Suggestions for Authors
I believe the authors have made very good revisions to the work, and it can now be accepted in its current form.
Author Response
I believe the authors have made very good revisions to the work, and it can now be accepted in its current form.
Response: Dear reviewer, thank you again for your professional comments on our manuscript, and sincerely wish you all the best.

Reviewer 2 Report
Comments and Suggestions for Authors
The authors have made efforts to improve the manuscript, but in their new text there are still errors, for instance: “Several studies have indicated that QUE exhibits effects against QUE can modulates cGAS-STING pathway to regulate macrophage polarization…”
In their response, the authors have clarified to the reviewer how they have selected the used dose, but they have not included this clarification in the paper. The dose justification should be available primarily for the readers, not(only) for the reviewer.
They have stated LSD as the post-hoc test, but this test is less conservative (has a higher type I error rate, i.e. is more prone to false positive results). A more conservative test (such as Tukey HSD) would be preferable.
With respect to Figure 6, we have zoomed the image to 400%, but the text is still unintelligible.
Comments on the Quality of English Language
The authors have made efforts to improve the text, but their new text seems to be affected by some new errors (or at least this is the impression given by the track changes version submitted).
Author Response
Dear Reviewer:
Thank you again for your professional comments on our manuscript and for giving us the opportunity to revise it. These professional comments will make our manuscript more substantial and professional. According to your comments, we have revised the manuscript once again. Some of your questions were answered below.
We hope that our manuscript is now suitable for publication in Microorganisms.
Manuscript ID number:
3160059
Title of paper:
Network Analysis of Gut Microbial Communities Reveal Key Reason for Quercetin Protects Against Colitis
The authors have made efforts to improve the manuscript, but in their new text there are still errors, for instance: “Several studies have indicated that QUE exhibits effects against QUE can modulates cGAS-STING pathway to regulate macrophage polarization…”
Response: Thank you for your suggestions. The content of the manuscript has been revised, and the erroneous sections have been corrected and highlighted in blue in the subsequently submitted manuscripts.
In their response, the authors have clarified to the reviewer how they have selected the used dose, but they have not included this clarification in the paper. The dose justification should be available primarily for the readers, not(only) for the reviewer.
Response: Thank you for your suggestions. We have supplemented it in the subsequently submitted manuscripts, and you can observed it in line 127 to 128.
They have stated LSD as the post-hoc test, but this test is less conservative (has a higher type I error rate, i.e. is more prone to false positive results). A more conservative test (such as Tukey HSD) would be preferable.
Response: Thank you for your suggestions. This study investigated and validated the regulatory effects of quercetin on inflammation, oxidative stress, and intestinal microbiota by employing a variety of indicators through multiple methodologies. Consequently, the analysis did not rely on a singular indicator to represent the findings. Therefore, we think that this analysis method is feasible for this study. If you still believe that our analytical approach is inadequate for this study, we are open to reanalyzing the data in future submitted manuscripts.
With respect to Figure 6, we have zoomed the image to 400%, but the text is still unintelligible.
Response: Thank you for your suggestions. We have made further adjustments to enhance the clarity of the figure 9. Additionally, the image included in the word version of the manuscript was clear; however, it became blurred following the system's automatic conversion to PDF format. To address this issue, we have resubmitted the original image for your review. We sincerely apologize for any inconvenience this may have caused.

Reviewer 3 Report
Comments and Suggestions for Authors
Dear author, I thank you for having considered the changes suggested in your manuscript, however, I consider it appropriate to send the document with the corresponding images because it is not clear which one will be part of the article.
On the other hand, I consider it appropriate to change the word “microbes” to “microbiota” in the conclusions.
It is suggested to restructure the caption of figure 9, because it contains repeated information, Figure 9 “Possible mechanism of QUE to treat the colitis in mice”. “QUE can effectively reduce symtoms in a mice model of DSS-induced colitis”.
I also consider it appropriate to describe the mechanisms that occur in each section comprising Figure 9, because the information is not clear.
Author Response
Dear Reviewer:
Thank you very much for your professional comments on our manuscript and for giving us the opportunity to resubmit a revised version of our manuscript. Your comments will make our manuscript more readable and professional. According with your advice, we amended the relevant part in manuscript. Some of your questions were answered below.
We hope that our manuscript is now suitable for publication in Microorganisms.
Manuscript ID number:
3160059
Title of paper:
Network Analysis of Gut Microbial Communities Reveal Key Reason for Quercetin Protects Against Colitis
Dear author, I thank you for having considered the changes suggested in your manuscript, however, I consider it appropriate to send the document with the corresponding images because it is not clear which one will be part of the article.
Response: Thank you for your suggestions. We have corrected it in the subsequently submitted manuscripts.
On the other hand, I consider it appropriate to change the word “microbes” to “microbiota” in the conclusions.
Response: Thank you for your suggestions. We have corrected it, and it can be observed in line 491 of the subsequently submitted manuscripts.
It is suggested to restructure the caption of figure 9, because it contains repeated information, Figure 9 “Possible mechanism of QUE to treat the colitis in mice”. “QUE can effectively reduce symtoms in a mice model of DSS-induced colitis”.
Response: Thank you for your suggestions. After our careful consideration, we have deleted part of the caption content of figure 9 because it is also repeated with the conclusion.
I also consider it appropriate to describe the mechanisms that occur in each section comprising Figure 9, because the information is not clear.
Response: Thank you for your suggestions. We have changed the content of figure 9 in the subsequently submitted manuscripts.

Round 3
Reviewer 3 Report
Comments and Suggestions for Authors
Dear authors, thank you for taking the observations made in the survey.
I wish you success in future publications
Author Response
Dear authors, thank you for taking the observations made in the survey.
I wish you success in future publications.
Response: Dear reviewer, Thank you again for your professional comments on our manuscript, your comments have made our manuscript more readable. We sincerely wish you all the best!
